

# Metabarcoding is (usually) more cost effective than seining or qPCR for detecting tidewater gobies and other estuarine fishes

Kevin Lafferty

U.S. Geological Survey Western Ecological Research Center, Santa Barbara, CA, USA

## ABSTRACT

Many studies have shown that environmental DNA (eDNA) sampling can be more sensitive than traditional sampling. For instance, past studies found a specific qPCR probe of a water sample is better than a seine for detecting the endangered northern tidewater goby, *Eucyclogobius newberryi*. Furthermore, a metabarcoding sample often detects more fish species than a seine detects. Less consideration has been given to sampling costs. To help managers choose the best sampling method for their budget, I estimated detectability and costs per sample to compare the cost effectiveness of seining, qPCR and metabarcoding for detecting endangered tidewater gobies as well as the associated estuarine fish community in California. Five samples were enough for eDNA methods to confidently detect tidewater gobies, whereas seining took twice as many samples. Fixed program costs can be high for qPCR and seining, whereas metabarcoding had high per-sample costs, which led to changes in relative cost-effectiveness with the number of locations sampled. Under some circumstances (multiple locations visited or an already validated assay), qPCR was a bit more cost effective than metabarcoding for detecting tidewater gobies. Under all assumptions, seining was the least cost-effective method for detecting tidewater gobies or other fishes. Metabarcoding was the most cost-effective sampling method for multiple species detection. Despite its advantages, metabarcoding has gaps in sequence databases, can yield vague results for some species, and can lead novices to serious errors. Seining remains the only way to rapidly assess densities, size distributions, and fine-scale spatial distributions.

## INTRODUCTION

According to *The Hollywood Reporter*, Chase Crawford of *Gossip Girl*, *NCIS* star Daniela Ruah, and *Justified's* Timothy Olyphant nearly had their weekend ruined by a tiny fish (*Gardner, 2023*). The endangered northern tidewater goby, *Eucyclogobius newberryi* (Girard, 1856) had been reported to be occupying a flooded underpass along the 2023 Malibu Triathlon cycling route and, if indeed present, would have perished if the area had been drained for the event. But the *Hollywood Reporter* is not known for accurate species distribution information, and the tidewater goby claims seemed a ploy intended to disrupt

Corresponding author
Kevin Lafferty, klafferty@usgs.gov

the event. Although a tidewater goby survey could have clarified matters, doing so would have been neither cheap nor easy. For these reasons, I compare the costs and benefits for various tidewater-goby survey methods.

The now infamous tidewater goby normally lives in shallow, brackish portions of coastal California streams, marshes, lagoons and estuaries between the Smith River to the north and Agua Hedionda Lagoon to the south (*Swift et al., 1989*). A close relative, the southern tidewater goby, *E. kristinae*, (*Swenson, 1999*), occurs south of the Palos Verdes Peninsula. Where found, tidewater gobies are often the most abundant fish species. Though small and drab, they have their charms. Notably, they tolerate seemingly intolerable conditions, and females compete for nests that males attend to (*Lafferty, Swenson & Swift, 1996*; *Swenson, 1999*). Since 1900, habitat loss and degradation combined with droughts led to extirpations, especially in southern California and San Francisco Bay (*Lafferty, Swenson & Swift, 1996*). To help conserve the species, the U.S. Fish and Wildlife Service (FWS) asks land managers to survey for tidewater gobies as part of habitat conservation plans or in response to construction projects. When a bridge needs retrofitting along the California coast, someone pays to determine if tidewater gobies are present or absent (*US Fish & Wildlife Service, 2005*).

Although it is not uncommon to confirm the presence of tidewater gobies in a single seine haul, seining for tidewater gobies requires a federal collection permit that can take several months to obtain and additional effort to maintain. For instance, the FWS expects that permitted individuals have at least 20 h of formal training in sampling methods and identification. This makes sense because tidewater gobies may be under-sampled with some methods (*e.g.*, traps, dipnets) and injured by others (*Lafferty, 2004*). Because seining can capture many fish species, one can quickly identify, count, and get size information for each haul. However, it takes expertise in identification to tell tidewater gobies from other co-occurring goby species with similar coloration, shapes, and sizes. To identify a tidewater goby, one inspects a fish immersed in a clear container for a vague clear patch at the tip of an erect first dorsal fin (*Lafferty, 2004*). Even with this technique, tidewater gobies can be difficult to distinguish from juvenile long-jaw mudsuckers (*Gillichthys mirabilis*). Another obstacle is that poor water quality, steep banks, snags, and vegetation can make seining impractical or unsafe. And sometimes tidewater gobies are rare enough that it takes many seine hauls to find them. In addition, while seining, it is difficult to avoid trampling goby burrows, leading to the ironic outcome that the main impact to a tidewater goby population is the day the biologists show up. Also, seining multiple sites has the potential to move biological material, and could inadvertently spread nuisance species or infectious diseases that could impact tidewater gobies or the community they live in. Thus, alternatives to seining have been sought.

Organisms scatter genetic material as they go about their lives and when they die. Using the polymerase chain reaction (PCR), even trace environmental DNA (eDNA) in environmental samples such as soil or water can be amplified to detectable quantities. After investing in developing a species-associated assay, qPCR can be as accurate as traditional ecological survey techniques for aquatic organisms (*Thomsen et al., 2012*; *Biggs et al., 2015b*; *Langlois et al., 2021*). And several studies have suggested that eDNA sampling can be more

sensitive at detection than traditional sampling (*Ficetola et al., 2008*; *Jerde et al., 2011*; *Dejean et al., 2012*). Indeed, Kinziger and colleagues pioneered sampling for tidewater gobies with qPCR (*Schmelzle & Kinziger, 2016*; *Sutter & Kinziger, 2018*; *Sutter & Kinziger, 2019*; *Martel et al., 2021*) and found detectability per qPCR sample was, on average 0.74 compared with 0.39 for a traditional seine (*Schmelzle & Kinziger, 2016*). Based on these promising results, some managers have expressed interest in qPCR-based sampling. For instance, the city of Santa Barbara, which had conducted many tidewater goby surveys over the years, found that qPCR sampling detected tidewater gobies at four of five sites in the Andrée Clark Bird Refuge with no amplification in negative controls (*Dressler et al., 2020*). *Dressler et al. (2020)* argued that although qPCR sampling takes time to process, and does not give densities or size information, it is permit free, takes less field time and expertise, is less environmentally impactful than seining, and is at least as accurate.

An increasingly popular alternative to qPCR is metabarcoding. Rather than amplify a species-specific assay, metabarcoding amplifies a variable gene region common to many taxa. High-throughput sequencing then amplifies unique sequence variants that can be classified by "blasting" against a sequence database to generate a list of similar sequences, each associated with reference species (*Deiner et al., 2017*). he allure of metabarcoding is that we can move closer to meeting *Elton's (1949)* long-held goal for ecology to describe multi-taxon communities, including small cryptic species like gobies (*Valentini et al., 2016*; *Bessey et al., 2020*). Indeed, metabarcoding often identifies more fish species than conventional methods like seining (*McElroy et al., 2020*). Tidewater gobies and several associated estuarine fishes are well-sampled by metabarcoding with 12S, 16S and cytochrome c oxidase subunit I (COI) primers (*Pfrender et al., 2017*). However, metabarcoding also has the potential to produce false-positive "samples" (due to contamination or database errors), false positive "sites" (correctly identified DNA that did not come from the site) and false negative occurrences due to limitations in sequence databases or sampling effort. Interpreting metabarcoding data is difficult, so such errors may not be detected by those lacking familiarity with a local system. For instance, *Pfrender et al. (2017)* considered non-estuarine species (*Sardinops sagax, Bodianus pulcher*) as present in estuaries based on detection with metabarcoding, whereas the exogenous DNA of these marine species were likely transported to the estuary in bird feces, by fishermen, or through currents. Metabarcoding is potentially less sensitive than qPCR for detecting individual species, especially when the target species is rare or not discernable from co-occurring close relatives. The few studies comparing qPCR and metabarcoding have found that metabarcoding is poorer than (*Wood et al., 2019*), close to (*Yu et al., 2022*), similar to (*Schneider et al., 2016*), or better than (*McCarthy et al., 2022*) qPCR for detecting a single species. In addition, of all the methods, metabarcoding is the most expensive on a per-sample basis. Therefore, metabarcoding would have to be quite effective at species detection to be cost effective.

Although several papers by eDNA practitioners have favorably compared eDNA sampling with conventional methods, most comparisons have not considered cost, which is often the key constraint for sampling programs. Formal cost-effectiveness analyses compare how rapidly effectiveness increases with the sum of fixed and variable investment (*Loomis & Walsh, 1997*). Although selecting monitoring methods based on cost-effectiveness

analysis has been desired (*Duffy, Luders & Ketschke, 1981*), it is rarely used in practice (*Caughlan & Oakley, 2001*). A particularly promising approach is to use modeling to compare cost-effectiveness under hypothetical scenarios (*Austin & Adomeit, 1991*; *Belbin & Austin, 1991*; *Lesser & Kalsbeek, 1997*). To answer a related question, *Stein et al. (2014)* found that molecular identification of separate specimens was more expensive than morphological identification, but that identification costs were comparable for molecular and morphological approaches for batch-specimen sequencing. It can cost surprisingly less to sample a site with eDNA than with traditional methods (*Biggs et al., 2015a*; *Sigsgaard et al., 2015*). Yet it was *Smart et al. (2016)* that first considered combining cost per sample and detection of European newt per sample as a way to compare the cost effectiveness of eDNA sampling and traditional sampling. They found that field sampling, though less effective per sample, was more cost effective due to its relatively low cost per sample. More recently, *Andres et al. (2023)* also considered cost per sample to compare different sampling methods, finding that eDNA was often a superior investment for detecting lake fish communities. Fixed costs (*i.e.,* assay development, lab set up, sampling gear costs) are less often considered than variable costs, as are environmental impacts that are not easily translated into time or money. Thus, more complex cost assessments seem merited.

With the aim to have greater return on investment when monitoring tidewater gobies, I conducted cost-effectiveness analyses that compared seining, qPCR and metabarcoding for monitoring tidewater gobies and estuarine fish communities. Although there are potentially many sampling goals (detection, time to information, density and size information), I focus on detection probability. Despite tidewater gobies and some methods being missing from some locations, there was enough existing and new information to make several pair-wise comparisons.

## MATERIALS & METHODS

### Field sampling

Water samples were taken for eDNA at three sites known to contain tidewater gobies in the past: 16 samples Calleguas Creek (34 06′42″, 119 04′54″, Ventura County), 21 samples at Ormond Lagoon (34 8′23″, 119 11′20″, just west of Naval Base Ventura County; and 17 samples at Santa Clara River Mouth (34 4′8″, 119 15′53″, Ventura County). Site access was authorized by NBVC - Point Mugu agreement number Q2 DAR-Q N6923222MP001XY (Fig. 1).

Sampling supplies and analyses were from Jonah Ventures ® and included metabarcoding kits (sampling supplies, DNA extraction, PCR, sequencing of six pooled PCR replicates, and bioinformatics) and qPCR kits (sampling supplies, DNA extraction, and qPCR of three laboratory replicates). An additional one-time cost was required to validate a previously published cytochrome *b* tidewater goby assay (*Schmelzle & Kinziger, 2016*). At all sites, nearshore water samples were taken wearing latex gloves to reduce contamination with human DNA. Samples were then filtered through a 1-micron disc filter by pushing water through the filter with a 60cc Luer-lock syringe until clogging (mean sample volume: 174 cc +/- 0.141 S.D.). Filter capsules were purged of water before

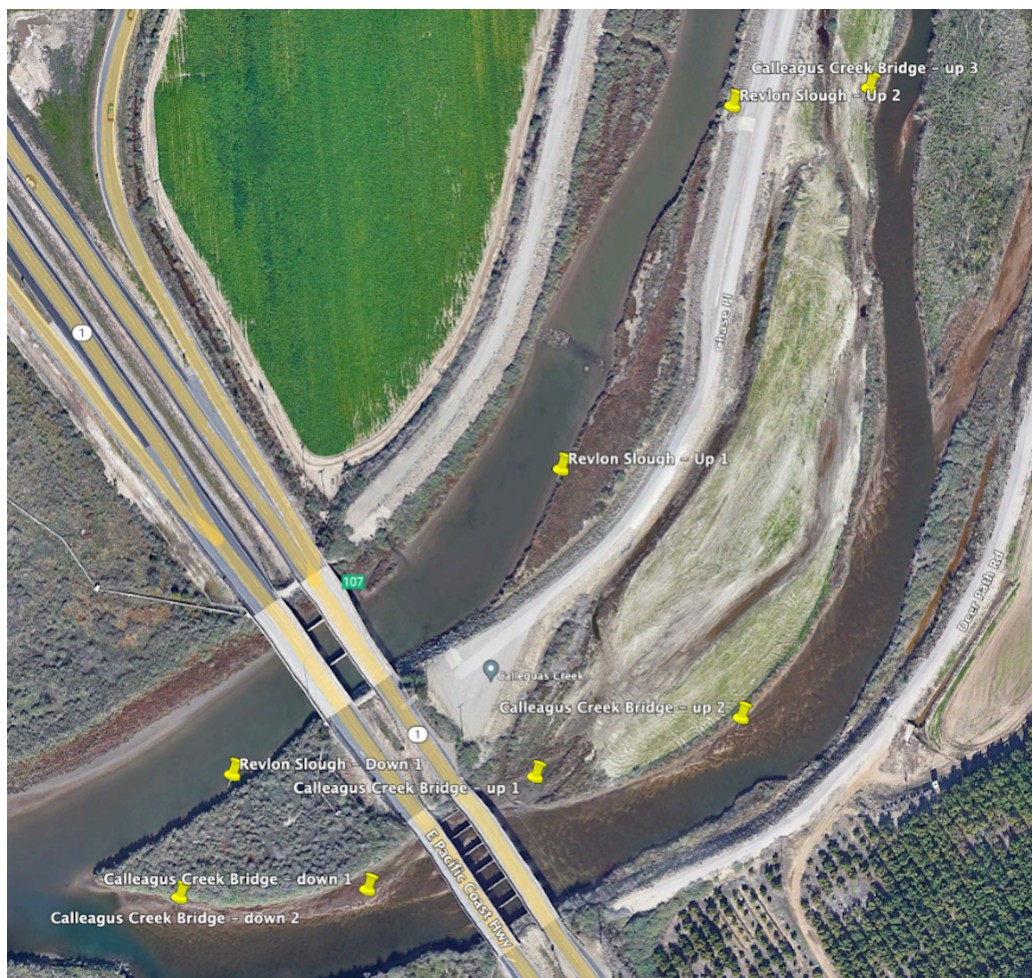

**Figure 1** **Eight sampling sites at the target locations around Calleguas Creek where it crosses Hwy 1 (Map data © 2023 Google).** Yellow pins indicate sampling sites. NBVC is located to the left of the Pacific Coast Highway in Oxnard. The north tributary is called Revlon Slough (north is up). See the Estuaries2022-MiFishU-metadata file (available at https://doi.org/10.25349/D9P60T) for latitude and longitudes of each site.

filling with preservative (tris-EDTA) then refrigerated until express shipped to Jonah Ventures ® for laboratory analysis.

To compare the efficacy of multi-species detection across methods, seines were taken at each sampling site for one of the estuaries (Calleguas Creek) that was sampled for eDNA (federal tidewater goby collection permit #PER0046428). Seine hauls were 2.4 m wide by 6.4 m distance on average in 0.6 m water depth. All captured fish were identified to species and released alive on site. Water temperature averaged 21.2 °C, conductivity was 32 mS (close to seawater), and dissolved oxygen was 8.3 mg/L.

## Lab methods

As part of the eDNA sample cost, DNA was extracted from the DNeasy Blood & Tissue Kit (QIAGEN©). Three qPCR replicates were run for the tidewater goby-specific cytB

assay (forward primer: 5′-CCTCAATTCTCGTTCTACTAGTTGT-3′, reverse primer: 5′-GAGAATAAGTACGTCTGCTACTAGG-3′, probe: FAM-ACGTGCACTGACCTTCC GGCCTTTCTCC-MGB). Metabarcoding was done for the mitochondrial 12S ribosomal RNA (rRNA) gene which was PCR amplified from each genomic DNA sample using the MiFishUF (5′-GTCGGTAAAACTCGTGCCAGC-3′) and MiFishU-R (5′-CATAGTGGGGGTATCTAATCCCAGTTTG-3′) primers with spacer regions. Amplicon size and PCR efficiency were visually inspected then cleaned by incubating. A second round of PCR was performed to complete the sequencing library construct. Final indexed amplicons from each sample were cleaned and normalized using SequalPrep Normalization Plates then pooled. Sample library pools were sent for sequencing on an Illumina NovaSeq 6000 (San Diego, CA, USA) at the Texas A&M Agrilife Genomics and Bioinformatics Sequencing Core facility (https://www.txgen.tamu.edu/) using the SP Reagent Kit v1.5 (500 cycles). Raw sequence data were then demultiplexed using pheniqs v2.10, primers were removed with Cutadapt v3.4, and read pairs were merged, denoised and chimeras were removed with VSEARCH v2.15.2. Amplicon sequence variants (ASVs) observed more than seven times were assigned taxonomy using a custom best-hits algorithm and a reference database that combined GenBank® and a Jonah Ventures® voucher sequence record, keeping any matching sequences with >90% agreement across top hits. Raw sequences were vouchered through the National Center for Biotechnology Information Sequence Read Archive (https://www.ncbi.nlm.nih.gov/sra; SRA# PRJNA924783).

## Fish community assessment with metabarcoding

To estimate fish community composition, I determined a consensus taxonomy from the several hypothesized taxa per amplicon sequence variant (ASV). ASVs with <8 total reads were removed due to low confidence and, for a given ASV, only those hypothesized taxa with the maximum percent match per ASV were retained. Hypothesized taxa meeting these criteria were refined using geography and habitat information compiled from information on fish communities previously seined at Calleguas Creek from four sources (most of which were gray-literature reports), as well as those species reported within 1° latitude or longitude of the study site from the Global Biodiversity Information Facility (*GBIF.org, 2024*) or the list of invasive species from the Global Introduced Species Database (*Poorter & Browne, 2005*). Assigned taxa not on this combined plausible list were considered highly implausible and were reassigned to their closest plausible relative (*e.g.*, *Fundulus lima* was re-assigned to the locally abundant, but unsequenced, *Fundulus parvipinnis*. The two are each other's closest relatives, and so genetic similarity is expected.) In cases where there was more than one plausible taxon per ASV, the consensus was the lowest common taxon. Two consensus taxa not associated with estuarine habitats were considered valid assignments, but from an exogenous source (*e.g.*, from bird feces).

## Modeling cost with effort

I then created a cost model from the individual cost parameters, resulting in the following equation that considers estimated total cost $H$ (sum of hour equivalents per location

sampled) has a fixed cost plus a variable cost:

$$fixed = \frac{P+G+A}{L} + B + WV + R$$
$$variable = k(v + n(tW + E + D + q))$$

I parameterized the cost model based on >30 years sampling tidewater gobies and estuarine fishes (Table 1). Specifically, I considered that it takes time to maintain a seining permit ($P$) and build a plausible species list from background knowledge ($B$). It also costs money for field gear ($G$). Fixed costs per location sampled were assumed divisible by the number of locations visited ($L$) per sampling program. Field effort was multiplied by the number of field workers ($W$) that visit the location (with a trained fisheries biologist's hourly rate equivalent to three field workers). Visiting a location ($V$) takes effort to prepare for as well as time and expenses for travel. At a location, one visits ($k$) sites. At each site, one takes ($n$) samples. Visiting a new site within a location takes time ($v$), as does taking a sample at each site ($t$) (seining takes a similar amount of time as collecting plus filtering a water sample). Seining can result in environmental impacts ($E$) related to incidental take and crushing burrows (this is the hardest cost to put into common units). At the lab, there may be costs to validating or developing an assay ($A$), depending on if the sequencer has validated the assay (no new cost), or if an assay has been published (moderate cost), or if a marker was not yet developed (large cost). Therefore, I investigate all three scenarios. There are also costs for sequencing each sample ($q$). For qPCR, there can be additional costs, per technical replicate ($r$), but these are now small enough ($5 USD) to be ignorable (I thus used the cost for three technical replicates per sample). Data entry and quality control ($D$) increases per site visit. The final cost was report writing ($R$), which, like data entry, is more labor intensive for multispecies than single species efforts. I expressed cost in the common currency of field-worker hours (assuming a 2023 technician's hourly rate of $30 USD).

## Modeling tidewater goby detection probability per sample by method

To estimate detection probability per sample for qPCR and seines, I used existing data for tidewater goby catch per unit effort (CPUE) and site-level detection obtained from 14 sites where tidewater gobies were present in a northern California survey (*Schmelzle & Kinziger, 2016*). To get conditional probabilities, I excluded sites where tidewater gobies were not detected. I also excluded the Virgin Creek and Pudding Creek sites, which only had three and five samples, respectively. I used the first three of *Schmelzle & Kinziger*'s (*2015*) six qPCR replicates to make the analysis comparable to typical studies. Detection per sample was coded as 0 (no detection) or 1 (detection), and estuary and sampling method (*e.g.*, qPCR and seine haul) were factors.

To estimate joint detection probability per sample for qPCR and metabarcoding, I used observations from qPCR and metabarcoding at the J-Street Canal, Ormond Lagoon (34 8′23″, 119 11′20″) and the Santa Clara River Mouth (34 4′8″, 119 15′53″) so that qPCR and metabarcoding were factors.

For each methods contrast, a logistic regression was used to estimate the estuary-level coefficient associated with detecting a tidewater goby in a sample for a particular method

**Table 1** **Estimated parameters for the cost model used to estimate the investment for each method (for sampling just gobies or the entire fish community).** Estimates are for a typical single location visit to Calleguas Creek. Units are technician hours or counts. Actual parameterization could vary based on the location being surveyed or lab being used.

| Investment | Variable | qPCR | Goby metabarcode | Fish metabarcode | Goby seine | Fish seine |
|---|---|---|---|---|---|---|
| Permit management | P | 0 | 0 | 0 | 15 | 15 |
| Background knowledge | B | 0 | 0 | 6 | 0 | 3 |
| Gear | G | 0.5 | 0.5 | 0.5 | 3 | 3 |
| Locations | L | 1 | 1 | 1 | 1 | 1 |
| Workers | W | 1 | 1 | 1 | 4 | 4 |
| Visit to location | V | 4 | 4 | 4 | 4 | 4 |
| k sites | k | 1:25 | 1:25 | 1:25 | 1:25 | 1:25 |
| Visit to site | v | 0.5 | 0.5 | 0.5 | 0.5 | 0.5 |
| Samples per site | n | 1 | 1,2 | 1,2 | 1 | 1 |
| Time per n | t | 0.3 | 0.3 | 0.3 | 0.3 | 0.3 |
| External per n | E | 0 | 0 | 0 | 3 | 3 |
| Assay development | A | 0, 8,85 | 0 | 0 | 0 | 0 |
| Sequencing per n | q | 1.8 | 3.5 | 3.5 | 0 | 0 |
| Data entry per n | D | 0.2 | 0.2 | 0.2 | 0.2 | 0.4 |
| Reporting per L | R | 3 | 3 | 8 | 3 | 6 |

where tidewater gobies were present. At sites where gobies are very easy to detect, any method would be suitable, whereas I was mostly interested in detecting gobies where they are rare. For that reason, rather than using mean detection to parameterize the cost-benefit model, I used the low-detection coefficient (and confidence intervals) from *Schmelzle & Kinziger*'s (*2016*) Ocean Ranch site (intercept (1.08) - 2.2 site effect) as a ''rare-case'' scenario. The Ocean Ranch coefficient was then transformed to a probability (using the plogis function in R) to parameterize the cost-benefit model. Detection per sample, $\gamma$, was thus used to calculate how the cumulative detection probability, $d$, increases with sampling effort (n*k).

$$d = (1 - (1 - \gamma)^{nk})$$

Then, to help guide method selection, I made three plots for each method: (1) detection probability ($d$) as a function of effort ($n$), (2) cost ($H$) as a function of effort ($n$), and (3) detection probability ($d$) as a function of cost ($H$). Because the costs of permitting and environmental effects associated with seining were uncertain, the cost plot was then reproduced for two additional assumptions about seining costs: no environmental costs, and no permitting or environmental costs. Also, given that fixed costs become relatively diminished as the number of locations sampled increases, I reproduced the cost plot for 10 locations (*e.g.*, the effort it took to sample the southern tidewater goby populations) and again with 100 locations (the latter encompassing all potential tidewater goby populations).

## Modeling species accumulation with effort

The number of species detected in a sample tends to increase with sampling effort to an asymptote. In addition, sampling effort can be applied at multiple scales. For instance, one

can take many samples at a site and/or sample many sites. Diversity should increase with both efforts, but due to spatial heterogeneity, one might expect samples from the same site to be more similar than samples from different sites. I simulated the expected species accumulation with effort by taking 1,000 random orderings of the samples for seining and metabarcoding at Calleguas Creek (one or two replicates per site, and 1 to N sites), then calculating the average cumulative richness as a function of sample size, to generate a curve that started at the average richness per sample and leveled off at total cumulative richness. To make the collector's curve more flexible for scenario modeling, I fit it using a multi-level model for species accumulation with effort using the Polce and Kunin method (*Kunin et al., 2018*), which can be represented as:

$$S = \frac{(a + k^z n^c)}{(b + n^c)}$$

where $S$ is the number of detected species, $k$ is the number of sites (or area sampled), $n$ is the number of samples per site, $z$ increases with beta diversity among sites (or the species–area relationship independent of effort), $c$ relates to the slope of the collector's curve within a site, and $a$ and $b$ are constants.

To compare the efficacy of metabarcoding and seining for fish communities, I used non-linear least squares (with starting values = 1 for all Polce and Kunin parameters) to fit median log(S) as a function of $n$ and $k$. Because metabarcoding had two samples per site, one of the samples was chosen at random to represent a site at each iteration. I then compared the predicted relationship between log(S) and $k$ between metabarcoding and seining. Then, by linking effort to cost using the investment model, I estimated the relation between, log(S) and investment. Two metabarcoding strategies were evaluated. Double barcode refers to taking two samples per site (as done in my field samples), whereas single barcode projects a single sample per site (as commonly done by others). To compare among methods for species detection I plotted species accumulation against effort, cost against effort, and detection against cost. As for tidewater goby detection, I made separate plots to explore assumptions about the cost of seining and the cost efficiency of sampling multiple locations.

Statistics and data analysis, and model fitting were performed in R version 3.6.3 (*R Core Team, 2020*), cleaned using Tidyverse (*Wickham et al., 2019*), and visualized using ggplot2 (*Wickham, Chang & Wickham, 2016*). Original data, supplementary statistical tables and R code are available as a U.S. Geological Survey data release (*Lafferty, 2024*). The R code makes it possible to reproduce the analyses and figures and explore how different cost models or parameters might affect the results.

# RESULTS

## Tidewater goby assessment

Tidewater gobies were visibly present during sampling at Ormond Lagoon. All 18 qPCR samples at Ormond Lagoon and 15 of 20 qPCR Santa Clara River samples were positive for tidewater gobies, as were all 18 metabarcoding samples at Ormond Lagoon and 18 of 20 metabarcoding samples at Santa Clara River. Indeed, the logistic regression found no

difference in detection between metabarcoding and qPCR at the Ormond Lagoon and the Santa Clara River Mouth sites ($p = 0.99$, Table S1). Thus, hereafter, I assumed detection by metabarcoding to be indistinguishable from qPCR. No tidewater gobies were detected by qPCR, metabarcoding or seining at the eight Calleguas Creek sites. For this reason, comparing the ability of seines and qPCR to sample tidewater goby used published data by *Schmelzle & Kinziger (2016)*. Consistent with *Schmelzle & Kinziger's (2016)* analysis of their data, detectability with qPCR was higher than with seining ($p < 0.0001$, Table S2). As described above, in the cost-benefit analysis, I used relatively low detection estimates from Ocean Ranch (qPCR and metabarcoding = 0.62, CI [0.42−0.78], *vs.* seining = 0.25, CI [0.12−0.43], Table S3). In general, there was more benefit to increasing sampling sites ($k$), than replicate samples within sites ($n$), so for the figures, $n$ is set to one sample per site.

## Cost effectiveness

Plotting detectability against sites sampled (Fig. 2, top panel) for Ocean Ranch showed that seining had less detectability than eDNA (especially at low density sites). However, eDNA and seining had nearly perfect detection for more than ten sites, even with relatively low detection probability. In other words, although eDNA had higher detection than seining, all methods achieved near-perfect detection under typical sampling efforts.

Costs increased with effort, but the different methods had different cost intercepts and slopes (Table 1, Fig. 2 middle panel). As an example, for a single sample at one location, metabarcoding cost 9 *h* and seining 42 *h*. Initial qPCR costs varied tremendously depending on whether an assay was validated or had to be developed. Indeed, having to develop assay from scratch made qPCR the least cost-effective method for a single location. Seining also had high fixed costs related to permitting and gear, and high environmental impacts per sample, though excluding permitting and environmental costs for seining did not make seining more cost effective than qPCR (Fig. 3). Metabarcoding was usually the most cost-effective method for detecting tidewater gobies due to its lower fixed costs than the other methods and its higher detection rate per sample than seining (Fig. 2 bottom panel). Exceptions occurred for cases where the sequencer had a specific tidewater goby marker in hand (true for future studies), or when many locations were visited (Fig. 4), in which case qPCR was slightly more cost-effective than metabarcoding.

## Fish community assessment
### Seines

Seines captured eight species at Calleguas Creek (Table 2), and a 9th (mullet, previously unreported) was seen during seining. Seining detected two previously unreported species for Calleguas Creek (*Quietula y-cauda* and *Syngnathus leptorhynchus,* two species known to occur in adjacent Mugu Lagoon (*Onuf & Quammen, 1983*). The lack of tidewater gobies at Calleguas Creek was consistent with the persistent marine conditions there, as evidence by the presence of marine molluscs (*Cerithideopsis californica*, *Magallana gigas*, and *Ostrea lurida*) that depend on tidal estuaries.

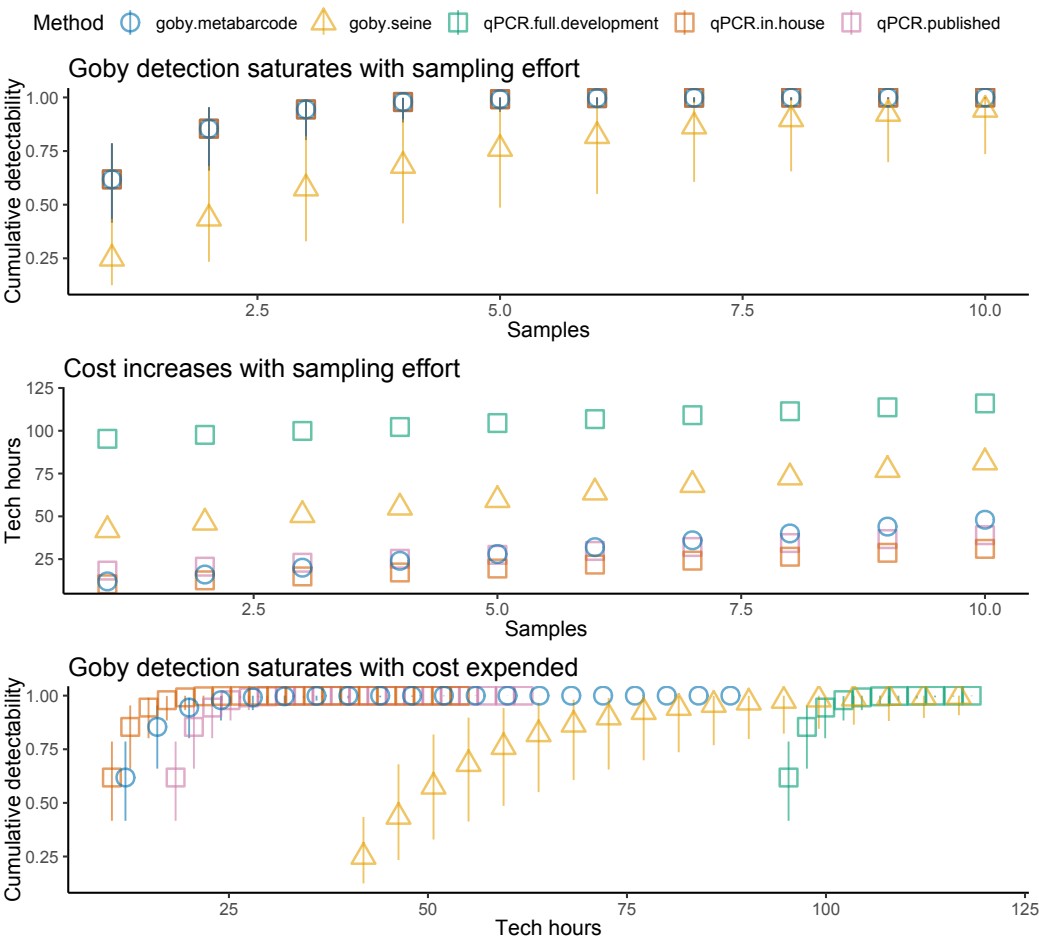

**Figure 2 Tidewater goby detection, cost, and cost effectiveness of seining, qPCR and metabarcoding.** Sampling effort (samples), sampling costs (hours per location), and tidewater goby detection (cumulative probability) for seining (triangles), qPCR (squares) and metabarcoding (circles). Bars represent 95% confidence limits. qPCR costs vary depending on whether the sequencer already has a validated assay (red = no cost), an assay is published but not validated (purple = moderate cost) or an assay needs to be developed from scratch (green = high cost).

## Environmental DNA

Not counting one ASV (12 reads) vaguely assigned to Leuciscidae (*e.g.*, *G. orcuttii* or *L. armatus*), metabarcoding detected DNA from 16 distinct fish species at Calleguas Creek (Table 2). Two of these were oceanic baitfish that seem most likely derived from bird feces. Of the remaining 14 estuarine fish species detected, 12 could be matched to a single species, and two were difficult to assign to the species rank. Specifically, although the Cypriniformes ASV was most likely *Carassius auratus, Cyprinus* was a plausible possibility too similar to exclude. Additionally, two gobies present in the system (*Quietula y-cauda* or *Ilypnus gilberti)* have not yet been sequenced and might be the source of the unidentified goby sequences. Metabarcoding detected to species all the fishes detected by seine except for *Syngnathus leptorhynchus* and, perhaps, *Quietula y-cauda*. Metabarcoding detected eight
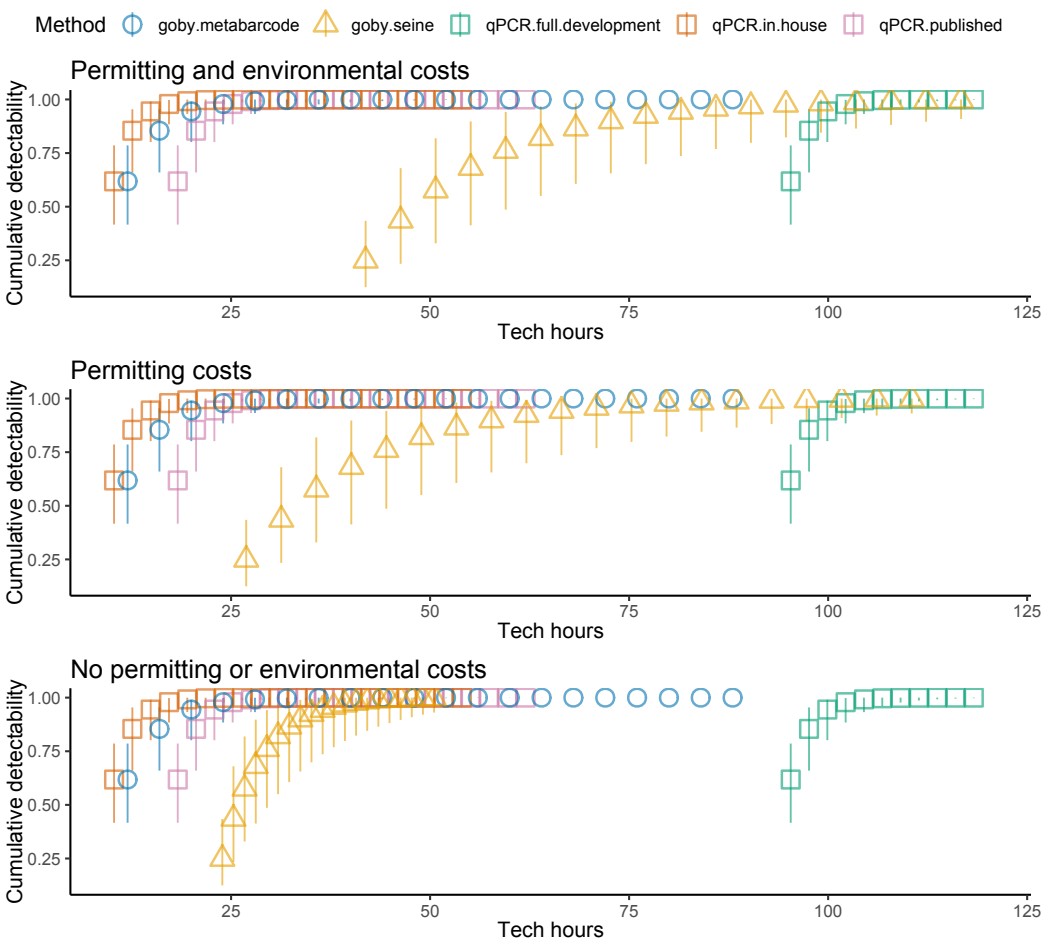

**Figure 3** **Fewer assumed seining costs reduces the estimated cost effectiveness of seining relative to environmental DNA (eDNA).** Tidewater goby detection (cumulative probability) *vs.* sampling costs (hours per location) for seining (triangles), qPCR (squares) and metabarcoding (circles). qPCR costs vary depending on whether the sequencer already has a validated assay (red = no cost), an assay is published but not validated (purple = moderate cost) or an assay needs to be developed from scratch (green = high cost). Bars represent 95% confidence limits. The top panel is the same as the bottom panel of Fig. 2. The middle (permitting costs, but no environmental costs) and lower panels relax assumptions about the costs of seining.

species not detected by seine. Moreover, it detected three previously unreported species for Calleguas Creek (*Citharichthys stigmaeus, Mugil cephalus,* and *Pleuronichthys guttulatus*). In addition to tidewater goby and the two unsequenced goby species, metabarcoding did not detect three previously reported freshwater fish species (*Ameiurus nebulosus, Cottus asper,* and *Lepomis cyanellus*). However, these species are detectable at other freshwater locations, indicating their DNA would have likely been detected had it occurred in the sample.
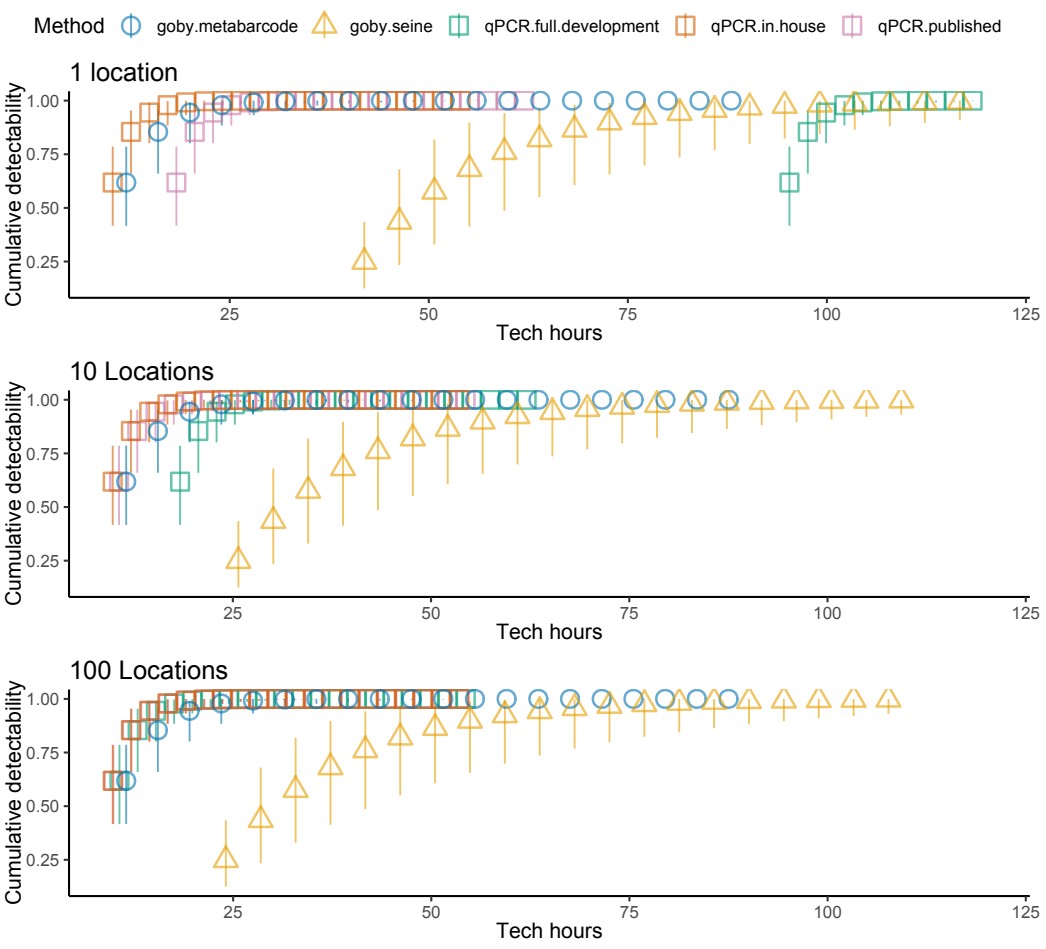

**Figure 4** **Sampling more locations increases the cost effectiveness of tidewater detection for seining and especially for qPCR.** Tidewater goby detection (cumulative probability) *vs.* sampling costs (hours) for seining (triangles), qPCR (squares) and metabarcoding (circles). qPCR costs vary depending on whether the sequencer already has a specific marker in hand (red = no cost), a specific marker is published but not in hand (red = moderate cost) or a specific marker needs to be developed from scratch (green = high cost). Bars represent 95% confidence limits. The top panel is the same as the bottom panel of Fig. 2. The middle and lower panels increase the number of locations visited.

## Species accumulation per sample by method

The Polce and Kunin species accumulation model fit to the qPCR and metabarcoding resampled accumulation curves made it possible to estimate parameters with error for incorporation into the methods comparison for species accumulation with effort (Table S4 for single sample, Table S5 for double sample).

## Cost effectiveness

Unsurprisingly, taking two metabarcoding samples per site detected more species than taking a single sample, but not twice as many species (Fig. 5). And taking two samples per site was not twice as expensive as taking one. Thus, overall, there was little difference in the cost effectiveness of taking one or two metabarcoding samples per site with respect to

**Table 2 Fishes detected at Calleguas Creek by different methods compared to previously published accounts.** Two species (Engraulis mordax, Leuresthes tenuis) are separated from the others as their exogenous DNA was probably transported in the feces of birds that have fed on coastal bait fishes. References: (1) KD Lafferty, 2014, unpublished data (Distribution of the endangered tidewater goby, Eucyclogobius newberry, and potential habitat, within the Mugu Lagoon and associated water bodies. U.S. Geological Survey Report to Naval Base Ventura County); (2) Bonterra Consulting, 2011, unpublished data (Results of the Tidewater Goby Survey for the State Route 1/Calleguas Creek Project Site Located West of Camarillo in Unincorporated Ventura County, California. Prepared for the USFWS, Ventura Office); (3) *Padre and Associates (2002)*, (4) KD Lafferty & RF Ambrose, 1996, unpublished data (Unpublished seine data in Ventura and Los Angeles Estuaries (1994–1996)); (5) *Swift et al. (1993)*.

| Species | Previous references | 2023 eDNA reads | 2023 Seine counts |
|---|---|---|---|
| *Ameiurus nebulosus* | 3 | 0 | |
| *Atherinops affinis* | 1,4,5 | 36,158 | 159 |
| *Carassius auratus* | 3 | ? | |
| *Citharichthys stigmaeus* | | 10 | |
| *Clevelandia ios* | 1,4,5 | 402 | 42 |
| *Cottus asper* | 2 | ? | |
| *Cymatogaster aggregata* | 8 | 752 | |
| *Cyprinus carpio* | 1 | ? | |
| *Eucyclogobius newberryi* | 1 | 0 | |
| *Fundulus parvipinnis* | 1 | 1,10,119 | 468 |
| *Gambusia affinis* | 1,3,5 | 304 | |
| *Gila orcuttii* | 3 | 0 | |
| *Gillichthys mirabilis* | 1,5 | 384 | 1 |
| *Hypsopsetta guttulata* | | 1,491 | |
| *Ilypnus gilberti* | 1 | ? | |
| *Lepomis cyanellus* | 5 | 0 | |
| *Leptocottus armatus* | 1,5 | 544 | 3 |
| *Micropterus salmoides* | 3 | 29 | |
| *Mugil cephalus* | | 19,013 | Visual |
| *Paralichthys californicus* | 1,4 | 28 | 6 |
| *Quietula y-cauda* | | ? | 1 |
| *Syngnathus leptorhynchus* | | 0 | 2 |
| Gobiidae (*Q. y-cauda* or *I. gilberti*) | | 176 | |
| Cypriniformes (*C. auratus* or *C. carpio*) | | 312 | |
| *Engraulis mordax* (exogenous) | | 14 | |
| *Leuresthes tenuis* (exogenous) | | 192 | |

detecting species. Nevertheless, given that taking two samples per site makes it possible to estimate within-site detection probabilities, taking two samples per site was the most informative strategy overall. Regardless, metabarcoding had higher species detection and lower costs than seining, and thus was much more cost effective. Assuming seining was less costly (Fig. 6) or several locations were visited (Fig. 7) made seining more cost effective, but not more cost effective than metabarcoding.

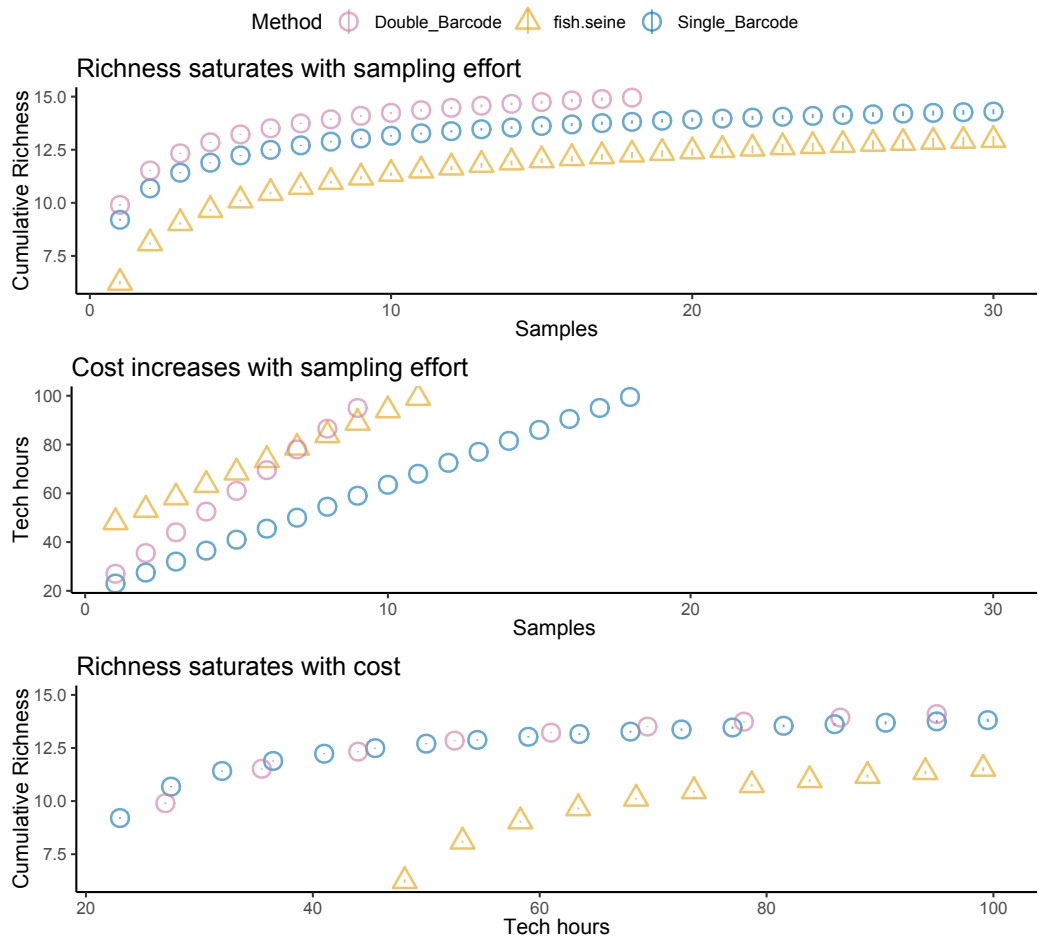

**Figure 5** **Fish detection, cost, and cost effectiveness of seining, qPCR and metabarcoding.** Sampling effort (n samples), sampling costs (hours per location), and species detection (cumulative richness) for metabarcoding (blue for one replicate, purple for two replicates), and seining (orange).

# DISCUSSION

For all methods, detection increased rapidly with sampling effort, so that even after just 3–5 samples, all methods would have confident tidewater goby detection at most sites, which is even better than the encouraging results found by *Dressler et al. (2020)*. Although a traditional beach seine is an effective way to detect tidewater gobies, eDNA sampling was even more cost effective.

Removing permitting and environmental costs of seining, or sampling more locations did not alter either conclusion. Thus, to confidently detect tidewater goby presence/absence at one location, the most cost-effective approach would usually be to take at least five water samples for eDNA analysis. For the data analyzed here, this would result in >99% chance of detection for ~25 technician hours. The most cost-effective eDNA method for detecting tidewater gobies depended on the assumptions. Validating or developing an assay would have made qPCR less cost effective than metabarcoding for a single location. But with an

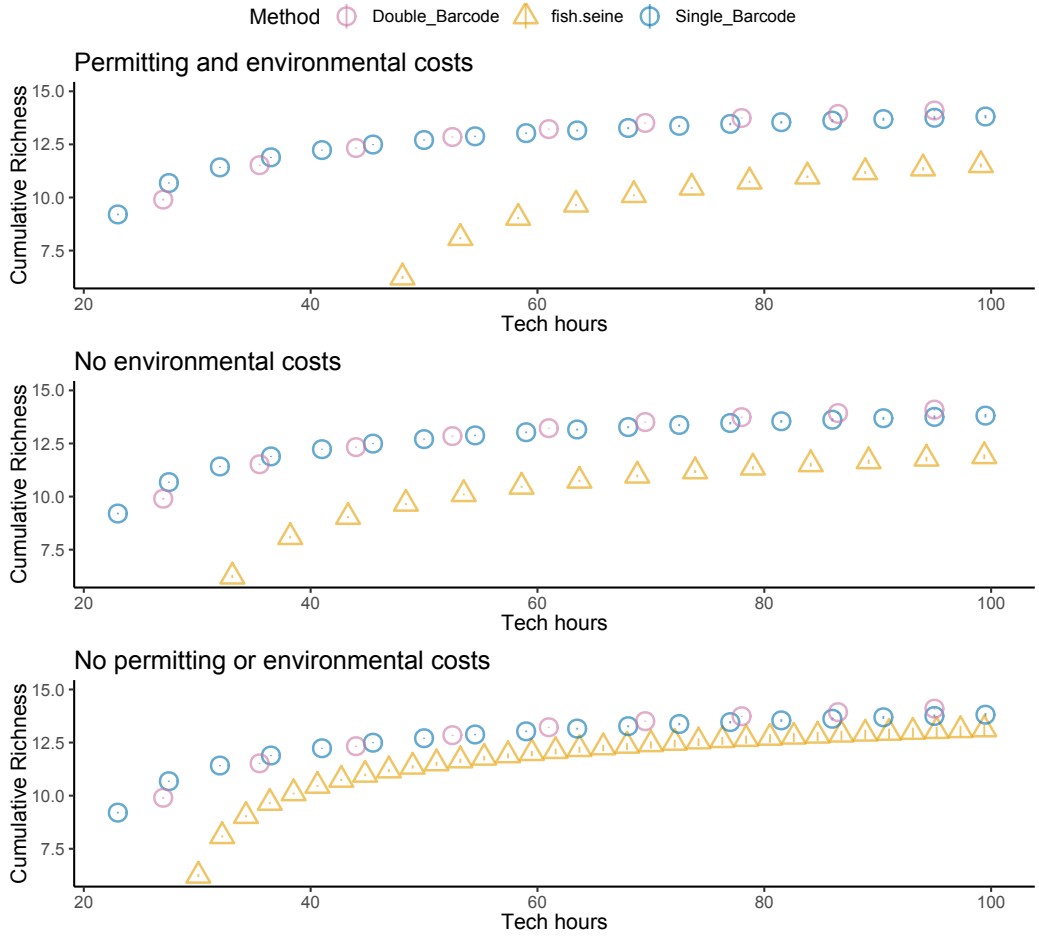

**Figure 6** **Fewer assumed seining costs does not make seining more cost effective than metabarcoding.** Species detection (cumulative richness) for metabarcoding (blue for one replicate, purple for two replicates), and seining (orange) *vs* costs (hours per location). Bars represent 95% confidence limits. The top panel is the same as the bottom panel of Fig. 5. The middle (permitting costs, but no environmental costs) and lower panels relax assumptions about the costs of seining.

assay now in hand, qPCR is slightly more cost-effective than metabarcoding. However, qPCR only indicates one species per assay, and, because each assay has a large, fixed cost, if managers want to know what other fishes live in an estuary, seining or metabarcoding would be the most cost-effective sampling methods.

For detecting the fish community, metabarcoding was more cost effective than seining. This result was also robust to removing permitting and environmental costs of seining. Roughly ten samples described most members of the fish community with metabarcoding Although metabarcoding is cost effective, it can be subject to false positives (*Jerde, 2021*). As such, novices ought not attempt to interpret metabarcoding community data without first carefully researching what fishes are likely to occur in a system. Also, currently a few species are not detectable with metabarcoding. These species should be identified and considered when reporting species lists determined by metabarcoding. If a manager

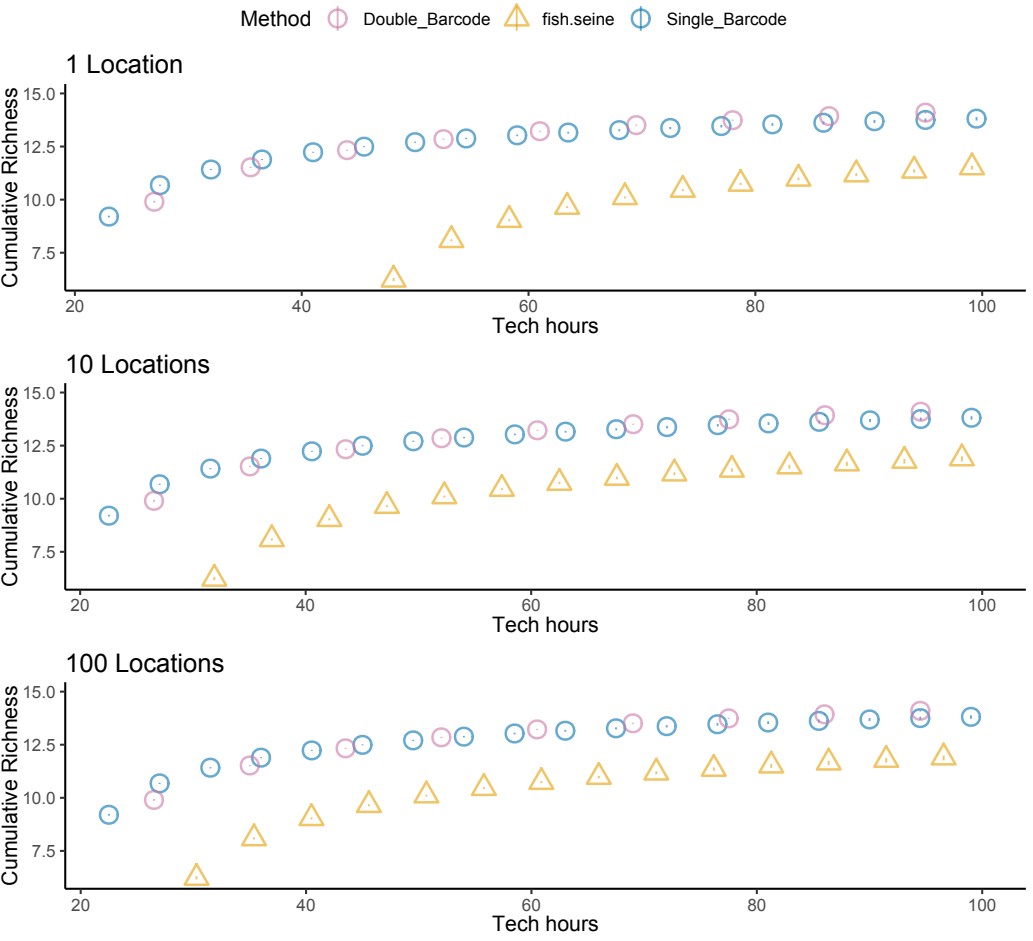

**Figure 7** **Sampling more locations does not make seining more cost effective than metabarcoding.** Species detection (cumulative richness) for metabarcoding (blue for one replicate, purple for two replicates), and seining (orange) *vs* costs (hours per location). Bars represent 95% confidence limits. The top panel is the same as the bottom panel of Fig. 5. The middle and lower panels increase the number of locations visited.

wishes to adopt metabarcoding for a fish-monitoring program, it might be prudent to first establish a list of species for which sequences are lacking, and then pay the fixed cost to sequence specimens for upload to to a genetic database (*e.g.*, GenBank®). This species "invisibility" is not unique to metabarcoding. Indeed, several species can evade seines by swimming, burrowing, or going to cover. Some of these species (like mullet) can be verified visually, and others enter baited traps. Indeed, all methods are subject to error and no single method captures all species, so using multiple methods can have the benefit of complementarity and confirmation. Confidently detecting the entire fish community may take traditional approaches to complement metabarcoding (*Andres et al., 2023*). In this system, complementary methods include beach seines, visual surveys (including snorkeling), trawls, trapping, burrow traps, and hook and line across replicated

sites, replicated visits, and different habitats. But if the goal is to create a core species list, metabarcoding is the single most cost-effective method.

Seining still has its advantages. If it is important to know goby abundance, density estimates can be obtained from many independent seine hauls with recorded dimensions. Although relative density can be estimated from qPCR (and to a lesser extent, metabarcoding) copy numbers, the association between reads and actual density is often moderate (*Lafferty et al., 2021*). Seining is also the most suitable method when managers want to compare fine-scale goby distributions within a system. This might be the case when a potential threat to tidewater gobies is localized (a bridge or culvert replacement) and the manager wants to know where to add protections. The spatial scale of a typical seine is finer than eDNA, which could easily come from distant locations, particularly when the system is subject to currents or tidal circulation. Although this is a disadvantage for eDNA sampling, this wider scale of inference is also likely why eDNA is often more sensitive than direct observation. Yet eDNA can be oversensitive; the presence of two exogenous fish species in the metabarcoding data indicate that tidewater goby DNA could be moved to a sample site by piscivorous birds or currents, leading to false-positive detections. A biologist might therefore question a first-ever detection of tidewater gobies at a new location (and especially a novel habitat) using eDNA without confirmation by seine. The biggest advantage of seining is perhaps that results are known in real time. If you have a triathlon scheduled, you might not be able to wait the days or weeks it currently takes to get back data from the sequencer. Although metabarcoding excelled at detection, seining is the most suitable method when managers want to rapidly know ecological densities, size frequency distributions, or health.

Knowing how many locations to sample and how far apart to space samples to adequately sample an estuary remains uncertain for eDNA sampling (*Yates et al., 2023*). Although the results suggest a rule of five for tidewater gobies, this might not apply to large systems or low-density tidewater goby locations. At such sites, the most cost-effective adaptive sampling program might be to collect more than five (*e.g.*, a dozen) samples, but only sequence all of them if the first five fail to detect tidewater goby DNA. A similar approach could be used for assessing the fish community with metabarcoding by collecting 20–50 samples, initially sequencing 10 of them, and then estimating how many additional samples to sequence to gain the desired community coverage.

This study has a limited taxonomic and geographic focus. Other fish species can be less distinguishable genetically or generate less detectable DNA. And, for many fish species, there could be a considerable (though ever decreasing) cost to developing a qPCR assay or obtaining reference sequences for metabarcoding. California estuaries are relatively well studied, and most fish species have been sequenced. Other parts of the world have received less attention, and this would likely create more sequence gaps in reference databases, leading to a higher probability of false negatives with metabarcoding. Also, the Mugu Lagoon system has been subject to many fisheries studies, which made creating a plausible fish list relatively easy. At sites with less background information, it would be harder to distinguish resident species from false positives. In addition, estuaries are distinct and relatively closed water bodies. Other marine habitats have considerably more potential for

eDNA to move across habitats. For instance, eDNA from intertidal sand flats at Palmyra Atoll was dominated by fish from adjacent reef habitat due to water movement and the relatively low diversity and abundance of the intertidal sand flat fish community (*Lafferty et al., 2021*). Although results from muddy estuaries in California might not generalize to coral reefs in the central Pacific, the value of considering costs when choosing methods apply to any system.

Environmental DNA methods create more sampling options than were available back when tidewater gobies were first listed in 1994. Indeed, seining, qPCR and metabarcoding are all good methods for detecting tidewater gobies. However, their cost effectiveness varies considerably within and among methods, depending on the details. Metabarcoding is an informative, environmentally friendly, and cost-effective way to measure presence-absence for one or many systems, and, given the declining cost of sequencing (*Metzker, 2010*), the cost-efficiency of eDNA sampling will likely increase over time. With an economic model like the one described here, managers can choose a method that best meets their information needs relative to their budgets.

## CONCLUSIONS

A qPCR sample is known to be better than seining at detecting tidewater gobies. Yet methods for detecting tidewater gobies and estuarine fishes have different costs. Applying a cost model and estimating detection probabilities as a function of effort revealed that metabarcoding was usually the most cost-effective method for detecting tidewater gobies and estuarine fishes in general. However, for subsequent survey efforts, qPCR may be a slightly more cost-effective method for sampling tidewater gobies. Given that metabarcoding has the added value of detecting the estuarine fish community, it was the most cost-effective method for detecting estuarine fishes. However, if a manager wants rapid results or information on fish demographics, seining is also a proven and reliable sampling method.

## ACKNOWLEDGEMENTS

Martin Ruane, Bill Hoyer, Francesca Ferrara, from NBVC and Zaine Zochair from UCSB assisted with field work. Nick Schulte and Joe Craine provided helpful comments on an early draft. Any use of trade, firm, or product names in this publication is for descriptive purposes only and does not imply endorsement by the U.S. Government.

### Funding

This work was completed under agreement number Q2 DAR-Q N6923222MP001XY as part of a study on Mugu Lagoon funded by the Naval Base Ventura County - Point Mugu, through Martin Ruane. Additional support came from the U.S. Geological Survey Ecosystems Mission Area. The funders had no role in study design, data collection and analysis, decision to publish, or preparation of the manuscript.

## Grant Disclosures

The following grant information was disclosed by the author:

Q2 DAR-Q N6923222MP001XY: Naval Base Ventura County - Point Mugu.

U.S. Ecosystems Mission area.

## Competing Interests

The authors declare there are no competing interests.

## Author Contributions

- Kevin Lafferty conceived and designed the experiments, performed the experiments, analyzed the data, prepared figures and/or tables, authored or reviewed drafts of the article, and approved the final draft.

## Animal Ethics

The following information was supplied relating to ethical approvals (i.e., approving body and any reference numbers):

US Fish and Wildlife, tidewater gobies collection permit #PER0046428

## Field Study Permissions

The following information was supplied relating to field study approvals (i.e., approving body and any reference numbers):

Site access was authorized by Naval Base Ventura County - Point Mugu

## DNA Deposition

The following information was supplied regarding the deposition of DNA sequences:

The raw sequences are available at NCBI SRA: PRJNA924783.

## Data Availability

The data and code are available at Dryad: Lafferty, Kevin (2024). Tidewater goby and estuarine fish records from seining, qPCR and metabarcoding data for Southern California estuaries in 2023 [Dataset]. Dryad. https://doi.org/10.25349/D9P60T.

## Supplemental Information

Supplemental information for this article can be found online at http://dx.doi.org/10.7717/peerj.16847#supplemental-information.

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
