# Peer review of "Metabarcoding is (usually) more cost effective than seining or qPCR for detecting tidewater gobies and other estuarine fishes"

_PeerJ, doi:10.7717/peerj.16847_

## Round 0.1 · original submission · Minor Revisions

Your manuscript has been reviewed and should be modified so that some points are clarified and others are much more precise. All of them have been indicated by the reviewers in their comments. Please, consider all these suggestions in the revised manuscript.
Best,

·

Basic reporting

1. Basic Reporting.
This paper is very good at outlining the state of the art and the need/importance of determining the methods of surveying and the trade-offs presented by the factors involved in sampling for this endangered fish, including nature and size of the habitat, sampling method or methods, time constraints as they relate to cost of sampling, including analysis of molecular samples and how these will change over time.

Experimental design

Design seems adequate with sample sizes and effort adequately accounted for. Someone more proficient in molecular and statistical methods is needed for those two issues, but they seem well done from my lesser familiarity. The only serious discrepancy I noted was on lines 142-145 says four sites sampled but only three are listed.
Could possibly suggest or comment on use of small plexiglass breeding tubes as a sampling method. Such tubes are quickly taken up when placed in proper environments during the extended breeding season, ie. the Swenson method. Easy to do and obviates the extended effort to snorkel. But requires at least two visits.

Validity of the findings

The findings seem very valid and is strikingly shown in the graphs of the extinction curves comparing the various methods and how they compare over time. This will be very valuable for investigators trying to balance time, cost, and objectives of individual studies. It also emphasizes the degree to which many aspects of the fish's biology and ecology are not elucidated despite the efficiency of being easily and cheaply detected or not.

Additional comments

4. General (and specific comments) comments
a. I wasn’t sure why the abstract was repeated twice at the beginning of the pdf.
b. Line 17, 41-42. Need to weave in mention of the fact that populations south of Palos Verdes peninsula were described as a distinct species, E. kristinae (Swift et al. 2016), the southern tidewater goby, recognized by Love and Passarelli (2020) and Page et al (2023). Thus, tidewater gobies occur almost throughout coastal California, but those studied here are northern tidewater goby.
c. Line 45, should also cite Swenson (1999) as the definitive paper on the ecology and behavior of northern tidewater goby.
d. Line 53-68, very good description of the issues and trade offs.
e. 101-104, several species could be detected in lagoons if they are partly open to the ocean, since rocky reefs often lie just offshore, collections of rocks and boulders washed down by high flow events. Would not need bird feces although these are sources also. The sheepshead, now in genus Bodianus, unlikely in bird diet.
f. Line 163-164. Is this a federal or state permit? Usually both are required or at least used to be, particularly for endangered species. Identified as a U. S. Fish and Wildlife Permit in Declarations but no mention of state permit.
g. Line 202-203, Fundulus lima and F. parvipinnis considered each other’s closest relative so this similarity is not surprising.
h. Lines 245-246; 309-310, 457, a variety of literature citation formats, presume this journal has some required, consistent format.
i. Line 349, spelling is “marker” rather than maker?
j. Line 356-57, Syngnathus leptorhinchus and Quietula y-cauda probably have been recorded before by Onuf and Quammen (1983).
k. 370-371, wording is confused here.
l. Diamond turbot now in Pleuonichthys guttulatus Girard.
m. 559-560, Should provide issue, number and pages for Hollywood Reporter. Several other gray literature reports could provide more details on how and where one might be able to see them if desirable.
n. Table 2: Engraulis and Leuresthes could be washed in with high nocturnal tides rather than with bird feces. Such Leuresthes become stressed in the fresher conditions in some lagoons and easy prey for terns. Ameiurus melas or A. natalis much more likely than A. nebulosus in most of coastal southern California (Swift et al. 1993) already cited. Cottus asper also surprising although possible as larvae coming down in the ocean and invading the lagoon, or from inland forms already known in the Santa Clara and Santa Margarita, and Santa Ana rivers, acquired as invasives via California aqueduct (Swift, et al. 1993 again).
o. The following are the complete references for my comments, not all of which need be included in the paper.

Love, M. S. and J. K. Passarelli. 2020. Miller and Lea’s guide to the coastal marine fishes of California. 2nd Edition, Davis, UC Agriculture and natural Resources Publication 3556.
Onuf, C. P. and M. L. Quammen. 1983. Fishes in a California coastal lagoon: effects of major storms on distribution and abundance. Mar. Ecol. Prog. Ser. 12:1-14.
Page, L. M. et a, 2023. Common and Scientific names of fishes from the United States, Canada, and Mexico. Eighth Edition. American Fisheries Society, Special Publication No. 37.
Swenson, R. O. 1999. The ecology, behavior, and conservation of the tidewater goby, Eucyclogobius newberryi. Environmental Biology of Fishes 55:99-119.
Swift, C. C., B. Spies, R. A. Ellingson, and D. K. Jacobs. 2016. A new species of the bay goby genus Eucyclogobius, endemic to southern California: evolution, conservation, and decline. PLoS One 11(7):e0158543.

My use of italics for all genus and species names apparently not carried over by copying into these review boxes.

·

Basic reporting

This is a good paper that needs some more work prior to publication

peerJ review -Lafferty costs

Metabarcoding is (usually) more cost effective than 2 seining or qPCR for detecting tidewater gobies and 3 other estuarine fishes

INTRODUCTION
35-44 This introduction to the introduction is based on what appears to be a false report of tidewater gobies in a location Where they have never been observed by any method, and where no study was conducted. I would lose this and improve the scientific setup for the interaction.

35-44 According to The Hollywood Reporter, Chase Crawford of Gossip Girl, NCIS star Daniela Ruah, 36 and Justifiedís Timothy Olyphant nearly had their weekend ruined by a tiny fish (Gardner, C., 37 2023). The endangered northern tidewater goby, Eucyclogobius newberryi (Girard, 1856) had 38 been detected in a flooded underpass along the 2023 Malibu Triathlon cycling route and would 39 have perished if the area had been drained for the event. But with a last-minute route change, the 40 celebrities went forward, and the fish did too. The now infamous tidewater goby normally lives 41 in shallow, brackish portions of coastal California streams, marshes, lagoons and estuaries 42 between the Smith River to the north and Agua Hedionda Lagoon to the south (Swift et al., 43 1989). Where found, they are often the most abundant fish species. Though small and drab, they 44 have their charms.

Overall the introduction is a little too loose it It seems to have been fun to write and is Pleasant to read. However it doesn't clarify some important points in the study. Namely that apples and oranges are being compared. Seining and barcoding detect and potentially quantify the abundance of multiple species. Where qpcr only to detects one species with some presumptive efficiency. There might also be some other benefits to seining such as determining size of fishes involved Etc. The molecular techniques do not do that. There are also other constraints and advantages to each of these techniques that might be mentioned in passing or shown in a table or something so managers could understand the benefit. For example “time to knowledge” of presence/ absence differs. if you find something with the a seine you know it's there right away (absence is another matter). qPCR will take hrs once the samples are in the lab. Barcoding is more laboratory and computationally intensive requiring more time and to some degree computational and taxonomic expertise. Thus in some ways this is undervaluing the expertise of the investigator in interpreting the Barcoding data.

The result that Barcoding is cost-effective for detection is interesting in large part because it has the ancillary benefit of recovering much of the other vertebrate fauna- although as noted it has some issues with false positives- in postmortem introduction - in that diversity.

Given Potential comparability of cost the ancillary benefits of different techniques should probably be explored to a greater degree. it should be emphasized that the cost comparisons relate specifically to detection and not these ancillary benefits. it is not that some of this is not mentioned it's just spread around especially in the introduction and so it may not be clear to the managerial reader what benefits really are accruing.



The following general knowledge/statement of problem should be conveyed in the introduction not just in the modeling part of the methods.

255-7 At sites where gobies are very easy to detect, any method would be suitable, whereas I was mostly interested in distinguishing sites where gobies are truly absent from sites where they are present, but rare (or otherwise difficult to detect).

More discussion of the different classes of data method and location used in the different analyses should be advertised in the intro and clarified in discussion. All data are not run in parallel - qPCR is from entirely different localities. Thus some clarity on why this approach is needed should be addressed before the middle of the methods i.e in the abstract and or introduction



Material and methods
136 -137I conducted cost-effectiveness analyses that compared seining, qPCR and metabarcoding for monitoring tidewater gobies and estuarine fish communities in general.

The separate quantification of value of species discovery only becomes apparent in the methods. That there are two separate objectives in terms of pres/abs and diversity and different methods to achieve them needs to be evident earlier in the abst & or intro

142 eDNA at four sites known to contain tidewater gobies in the past:
Elsewhere (in figures) this is listed as though gobies were known to be present at these locations- not that they were just recorded there in the past. check this relative to figures etc


366- 367 Two of these were oceanic baitfish that seem most likely derived from bird feces. Of the remaining 14 estuarine fish species detected, 12 could be matched to a single species, and two were difficult to assign to the species rank.
The above is an important pt re eDNA and detection via whatever method. However I would say this with less certainty as these fish may enter the larger Lagoon.

Experimental design

Not relavant

tests seem adequate and appear to be appropriate stats wise However, greater clarity of the different data sources and tests used needs to be clarified and could make i htis a very strong paper.

Validity of the findings

This is a very useful set of work that should be well received.

Not sure what data can be provided.

Discussion and conclusions could be tightened

---

## Round 0.2 · accepted · Accept

Thank you very much for sending us the manuscript of your article again. I revised again the manuscript and it can be seen that it has been improved and modified taking into account all the reviewers' considerations. Therefore, I am pleased to confirm that your paper has been accepted for publication in PeerJ.